# The Impact of Quality of Public Administration on Local Economic Growth in Vietnam

Thanh Hung Pham [1], Thi Thanh Hang Hoang [2], Eleftherios I. Thalassinos [3,4] and Hoang Anh Le [5,*]

[1] Faculty of Postgraduate Education, Banking University HCMC, No. 36 Ton That Dam Street, Nguyen Thai Binh Ward, District 1, Ho Chi Minh City 700000, Vietnam; thanhhungstcbp@gmail.com
[2] Department of Academic Affairs, Banking University HCMC, No. 36 Ton That Dam Street, Nguyen Thai Binh Ward, District 1, Ho Chi Minh City 700000, Vietnam; hanghtt@buh.edu.vn
[3] Faculty of Maritime and Industrial Studies, University of Piraeus, 185-33 Piraeus, Greece; thalassinos@ersj.eu
[4] Faculty of Economics, Management and Accountancy, University of Malta, 2080 Msida, Malta
[5] Institute for Research Science and Banking Technology, Banking University HCMC, No. 36 Ton That Dam Street, Nguyen Thai Binh Ward, District 1, Ho Chi Minh City 700000, Vietnam
* Correspondence: anhlh_vnc@buh.edu.vn

**Abstract:** This study examines how the quality of public administration influenced local economic growth in Vietnam from 2011 to 2019. Based on previous studies, we evaluate this impact through the Cobb–Douglas function includes government capital, thereby examining both the individual and interactive effects of local government expenditures and quality of public administration on local economic growth in Vietnam. The system GMM method (SGMM) was used to estimate the model with data collected from 61 provinces and cities in Vietnam in the period 2011–2019. The findings suggest that local government expenditures and quality of public administration positively influence local economic growth in Vietnam. Thereby, the authors propose policy implications to improve the efficiency of local government expenditures on local economic growth in Vietnam in terms of public administration.

**Keywords:** quality of public administration; local government expenditures; economic growth; PAPI index; SGMM

## 1. Introduction

Economic growth is the most important macroeconomic variable that reflects the overall performance of each country due to the production of more goods and services, improved productivity, and growth in the supply of labor. Increased productivity involves a combination of the workforce, physical capital such as plant and equipment, and increased use of new technology.

Previously, Vietnam was among the poorest countries in the world, but now, in the East Asia Pacific region, Vietnam has become one of the most dynamic countries. From 2002 to 2018, according to World Bank (2020), more than 45 million people escaped from poor and near-poor households, with the rate from over 70% to less than 6%. Currently, the GDP per capita of Vietnam increased to USD 2700 and by 2.7 times in 2019.

However, Vietnam still has many limitations that need to be assessed and considered, in which an important issue is in public spending and quality of public administration, in particular: (i) effective management of expenditure with scale still scattered and lack of concentration; (ii) the planning to allocate local government expenditure estimates has not been considered based on output results; (iii) the public administration performance indicators are still heavily emotional.

From the 1990s onwards, the role of quality of public administration in local economic growth has received more and more attention.

Researchers began to theorize about the relationship between the quality of public administration and economic growth (Fukuyama 1995; Acemoglu et al. 2004; Mishkin

2001). Further, empirical studies also showed evidence of the impact of the quality of public administration on economic growth, supporting the theory. Most of those studies have shown a positive impact of the quality of public administration on economic growth. One of them is the study by Kaufmann and Kraay (2004), which demonstrated that good public administration leads to higher per capita income. Furthermore, Grindle (2004) emphasized that the basic aim of good public administration is to eliminate poverty, combat corruption, and address other impediments to economic growth.

In addition, a few previous studies showed that the quality of public administration has had an impact and created many changes in the relationship between local government expenditure and economic growth (Glaeser and Saks 2004; Le et al. 2020). Specifically, the quality of public administration was considered a catalyst and good control while enhancing efficiency in estimating and using expenditures appropriately, thereby promoting economic development (Siddiqui and Ahmed 2013).

Although there is evidence that the quality of public administration has positively influenced economic growth, there are no studies that comprehensively examine how the components of public administration influence local economic growth. In addition, to the best of our knowledge, no study on the impact of public administration quality on economic growth has been conducted at the local scale in Vietnam. Therefore, this study is conducted to fill this research gap and propose appropriate policy implications. Our study aims to answer the following research questions:

Does good public administration quality really increase local economic growth in Vietnam?

Do components of public administration affect local economic growth in Vietnam?

The research model is built through the Cobb–Douglas production function to accomplish the above goal. Based on the studies of Siddiqui and Ahmed (2013), Cooray (2009), and Alexiou (2009), we use data collected from 61 provinces and cities in Vietnam in the period 2011–2019. By employing the System Generalized Method of Moments (SGMM) method, we estimate the model to examine how the components of public administration influence local economic growth. The Viet Nam Provincial Governance and Public Administration Performance Index (PAPI Index) is also included in the research model to represent the quality of public administration with data from the Vietnam Union of Science and Technology Associations and the United Nations Development Program (UNDP).

## 2. Literature Review

### 2.1. Definition of Concepts

**The quality of public administration**

The simplest definition of public administration is a set of values, policies, and institutions through which a society manages its economic, political, and social affairs. Public administration is conducted through the relationship between government, civil society, and the private sector. It includes mechanisms and processes for citizens and society to gain benefits, settle conflicts, and exercise their legal rights and responsibilities. Therefore, the quality of public administration is an expression of the quality of rules, institutions, and mechanisms that constrain and provide incentives for individuals, organizations, and corporations (Lok and Crawford 1999).

**Local Economic Growth**

Local Economic Growth is measured by the growth of Gross Regional Domestic Product (GRDP) within a region in each country (Feriyanto et al. 2020). According to the neoclassical theory, economic growth is measured by total output, a function that depends on the accumulation of capital, labor, and technological progress. In addition, over time, many studies have discovered other factors that have an impact on economic growth, such as human capital, government, or institutional size in some countries (Barro 1990; Barro and Sala-i-Martin 2004).

## 2.2. Theories Related to the Impact of Public Governance Quality on Economic Growth

**Theory of public administration**

Smith (1755) argues that the prerequisites for building the most prosperous state from the least prosperous state are peace, taxes, and an accepted government of justice. According to Buchanan (1987), it has also been argued that economists should look to the constitution of the economic institution itself to examine the rules and constraints in which political actors play an important role.

**The theory of public choice and theory of political economy**

Dethier (1999) states that governments are not benevolent rulers who strive to maximize social benefits, but the interrelationships between public authorities complicate public governance structure.

In in-depth analysis of political economic theory on government expenditure, Larcinese et al. (2010) explained the impact of quality of public administration on the relationship between government expenditure and economic growth through three hypotheses about the distribution policy of the state budget of a country: the "swing-voter" hypothesis, the "electoral battleground" hypothesis, and the "partisan supporters" hypothesis.

**New institutional economic theory**

The role of public administration in growth theory can now be viewed as part of a line of theories about the "determining factors" of economic growth and development (Zhuang et al. 2010).

Totikidis et al. (2005) argue that public administration programs supported by the United Nations have helped increase accountability and transparency in local and national public administration practices.

With good public administration, the government can prevent interest groups from seeking profits from arbitrary exploitation of power (Buchanan 1987). Barro (1991) shows a negative association between political violence and private investment.

## 2.3. Evidence on the Impact of Quality of Public Administration on Economic Growth

The neoclassical growth model is considered the first standard model, converging quite a few factors determining economic growth in the long run. However, the neoclassical growth model has been both a huge success and a failure. A major limitation of the neoclassical growth model is that factors affecting economic growth, such as labor efficiency, are determined exogenously in the long run.

The limitations of the neoclassical growth model have prompted many research directions to expand the model to better suit the realities of developing countries, and have led to the birth of endogenous growth models. In particular, productivity gains that result from accumulating human capital or patenting activities are the drivers of long-term growth. Endogenous growth models, especially those that take into account human capital, have contributed to explaining income disparities across countries. Despite their significant contributions, endogenous growth models still suffer from some limitations because they ignore growth-restricting factors such as the quality of public administration in developing countries.

Although the detailed study of the role of quality of public administration in economic growth is relatively new, the importance of good public administration has been recognized since the 18th century, as demonstrated by the theory of Smith (1755). He argues that the prerequisites for building a state of highest prosperity from a state of least prosperity were peace, taxes, and an accepted administration of justice.

Besides the theory of Smith (1755), the importance of quality of public administration is also confirmed through the theory of Buchanan (1987). He argues that economists should look to the constitution of the economic institution itself to examine the rules and constraints in which political actors play an important role.

Although the theories of Smith (1755) and Buchanan (1987) have shown the importance of public administration, the role of public administration in economic activity was really highlighted only when the study of Ndulu and O'Connell (1999) was published. They find that dictatorship is associated with a weak economy. Good public administration enables citizens to participate in politics and activities related to empowerment, which in turn can improve economic performance and promote growth.

Dethier (1999) argues that the effective use of public resources depends on incentive programs of public organizations that focus on ensuring commitment and policy implementation to maximize social welfare. Good public administration will improve human capital and improve efficiency in the use of resources, thereby enhancing economic growth (Dethier 1999).

However, the political economy theory also holds that those in power conceive public administration to transfer resources to themselves (Acemoglu et al. 2004). Acemoglu et al. (2004) argue that different interest groups will prefer different institutions and that groups with stronger political power will ultimately determine the establishment of governance. This will affect economic growth because the governance system is set up to protect the interests of an interest group. Political institutions establish a legal system that prescribes the rules for controlling volatility. Therefore, different interest groups will compete for political power and economic rent within the legal system's rules. Without an appropriate incentive mechanism in political institutions, implicit rules can be established to benefit particularly politically advantageous groups. This will affect the country's economic growth due to conflicts of interest. On the other hand, without the basic legal protection of ownership by the government for individuals, the growth rate of private investment will be slowed down, reducing the economic growth rate. Private investment, especially foreign investment, is also discouraged because of the influence of bureaucracy. This will also slow down economic growth. In a more specific perspective, Faruq (2011) argues that poor public administration, as demonstrated by rampant corruption, bureaucracy, and high risk of private property being confiscated by the government, can cause manufacturers to not make investments and innovations in the long run.

## 3. Methodology

We begin the methodology section with the construction of a model to assess the impact of public administration quality on local economic growth in Vietnam. Then, we show how to collect data for the variables in the model. At the same time, we also present the reasons for choosing the SGMM method to estimate the model as well as the necessary tests.

### 3.1. Research Model

Unlike previous studies, which often build models based on the synthesis of variables that have an impact on economic growth, in this study, we perform mathematical transformations from the Cobb–Douglas production function. This approach helps to make the model built more convincing. Furthermore, with this approach, our study can compare the impact of public administration quality on local economic growth in an Eastern country, Vietnam, with previous results from Western countries.

We start with a production function of the following form:

$$GRDP_{it} = A(K_{it})^{\beta_1}(L_{it})^{\beta_2}(G_{it})^{1-\beta_1-\beta_2} \tag{1}$$

where $GRDP_{it}$ is the gross regional domestic product, $A$ is the technological progress, $G_{it}$ is local government expenditure, $K_{it}$ is private investment capital of the province, and $L_{it}$ is the province's labor force (with $\beta_1 + \beta_2 < 1$).

Then, we take the natural logarithm of the two sides of (1), and we obtain:

$$lnGRDP_{it} = \beta_0 + \beta_1 lnK_{it} + \beta_2 lnL_{it} + \beta_3 lnG_{it} + \varepsilon_{it} \tag{2}$$

where *t* represents the *t*-th year, and *i* represents the *i*-th province.

To display the local economic growth, we continue to minus both sides of Equation (2) for $lnGRDP_{it-1}$, and the model obtained is as follows:

$$lnGRDP_{it} - lnGRDP_{it-1} = \beta_0 + (\rho - 1)lnGRDP_{it-1} + \beta_1 lnK_{it} + \beta_2 lnL_{it} + \beta_3 lnG_{it} + \varepsilon_{it} \tag{3}$$

Thereby, the local economic growth is calculated as follows:

$$growth_{it} = lnGRDP_{it} - lnGRDP_{it-1}$$

To assess the impact of quality of public administration on local economic growth in Vietnam, we adjust Equation (3) based on the studies of Cooray (2009) and Siddiqui and Ahmed (2013). Specifically, the indicators of the public administration and governance performance index (PAPI) are included in Equation (3) to represent the quality of public administration.

$$growth_{it} = \beta_0 + (\rho - 1)lnGRDP_{it-1} + \beta_1 lnK_{it} + \beta_2 lnL_{it} + \beta_3 lnG_{it} + \beta_4 PAPI_{it} + \varepsilon_{it} \tag{4}$$

The study was conducted in the period 2011–2019 to examine the impact of quality of public administration on local economic growth in Vietnam. The study is statistically in 61 provinces and cities in Vietnam and has ensured over 90% of the localities in Vietnam. Table 1 below describes the variables in the model, how they were calculated, and the data source.

**Table 1.** Summary of variables in the model.

| ID | Variables | Notation | Measures | Data Sources |
|---|---|---|---|---|
| 1 | Economic growth | $growth_{it}$ | $lnGRDP_{it} - lnGRDP_{it-1}$ | General Statistics Office of Vietnam, the Statistical Office of 61 provinces and cities in the sample |
| 2 | GRDP at the beginning of the period | $lnGRDP_{it-1}$ | Natural logarithm of GRDP in year *t* 1 | |
| 3 | Investment capital | $lnK_{it}$ | Natural logarithm of the private investment capital of the province. | |
| 4 | Labor force | $lnL_{it}$ | Natural logarithm of the province's labor force. | |
| 5 | Local government expenditure | $lnG_{it}$ | Natural logarithm of the local government expenditure. | |
| 6 | Quality of public administration | $PAPI_{it}$ | PAPI index of provinces and cities i in year *t* | The Vietnam Union of Science and Technology Associations and the United Nations Development Program (UNDP) in Vietnam |
| 6.1 | Participation of the people at the grassroots level | $TG_{it}$ | TG ingredient index of provinces and cities i in year *t* | |
| 6.2 | Transparency | $CK_{it}$ | CK ingredient index of provinces and cities i in year *t* | |
| 6.3 | Accountability to the people | $TN_{it}$ | TN ingredient index of provinces and cities i in year *t* | |

| ID | Variables | Notation | Measures | Data Sources |
|---|---|---|---|---|
| 6.4 | Control corruption in the public sector | $KS_{it}$ | KS ingredient index of provinces and cities i in year $t$ | |
| 6.5 | Public administrative procedures | $TT_{it}$ | TT ingredient index of provinces and cities i in year $t$ | |
| 6.6 | Public service provision | $CU_{it}$ | CU ingredient index of provinces and cities i in year $t$ | |

*3.2. Estimation Method and Data*

This study employed the SGMM method of Arellano and Bond (1991) for estimating model 4. The SGMM estimation is often used in panel data in the case of autocorrelation and heteroskedasticity. At the same time, through this method, the endogeneity phenomenon, which often occurs in macroeconomic models (Konstantakopoulou 2022), is overcome.

There are several reasons to use the SGMM method. Firstly, the panel data of the study have a small time series for each province (9 years) and a large number of provinces (61 provinces), which means a smaller timeline but lots of observations. Secondly, model 4 contains the independent variable which is the lagged variable of the dependent variable. Thirdly, the SGMM method can be estimated when the independent variables are not strictly exogenous. Finally, when the model contains autocorrelation and heteroskedasticity, this method is suitable.

Besides, we also perform tests to ensure the model is reliable.

**Test of residual autocorrelation**. According to Arellano and Bond (1991), the GMM estimate requires a first-order correlation and no second-order correlation of the residuals. Therefore, when testing hypothesis H0: there is no first-order correlation (AR(1) test) and no second-order correlation of residuals (AR(2) test), we reject H0 in the AR(1) test and accept H0 in the AR(2) test.

**Check the fit of the model and representative variables**. Similar to other models, the fit of the model can be inspected through the F-test. In addition, the Sargan/Hansen test is also used to test the hypothesis H0: the instrumental variables are suitable. Accepting hypothesis H0 means that the instrumental variables used in the model are appropriate.

## 4. Empirical Result

*4.1. Descriptive Statistics*

Table 2 presents the descriptive statistics of the variables in the model.

**Table 2.** Descriptive statistics of variables.

| Variables | No. of Observations | Mean | Standard Deviation | Min | Max |
|---|---|---|---|---|---|
| GRDP | 549 | 82,373.490 | 149,289.600 | 4073.500 | 1,344,743 |
| G | 549 | 18,309.130 | 20,054.680 | 3690.300 | 287,857.300 |
| K | 549 | 32,384.640 | 56,439.850 | 3018.261 | 561,437.700 |
| L | 549 | 59.833 | 3.861 | 50 | 72.994 |
| TG | 549 | 5.198 | 0.493 | 3.751 | 6.809 |
| CK | 549 | 5.601 | 0.505 | 4.435 | 7.240 |
| TN | 549 | 5.358 | 0.567 | 4.097 | 7.506 |
| KS | 549 | 6.117 | 0.658 | 4.054 | 8.190 |
| TT | 549 | 7.057 | 0.329 | 5.895 | 7.947 |
| CU | 549 | 6.982 | 0.362 | 5.681 | 8.028 |

The results of descriptive statistics show that the average GRDP of provinces in Vietnam in the period 2011–2019 was VND 82,373.49 billion. The standard deviation of GRDP is 149,289.6 billion, showing that the difference in GRDP of localities in Vietnam is

quite large. This large fluctuation shows that the development of localities in Vietnam is not uniform.

For Citizen Participation (TG), the mean of this variable is 5.20 points, and the standard deviation is 0.49 points, indicating a relatively low level of volatility in TG. For Publicity and Transparency in Decision Making (CK), the mean of this variable is 5.60 points, and the standard deviation is 0.50 points, indicating that the volatility of CK is relatively low. For Citizens Accountability (TN), the mean of this variable is 5.36 points, and the standard deviation is 0.57 points, indicating a relatively low level of TN volatility. For the Control of Corruption in the Public Sector (KS), the mean of this variable is 6.12 points, and the standard deviation is 0.66 points, indicating a relatively low degree of KS volatility. For Public Administration Procedures (TT), the mean value of this variable is 7.06 points, and the standard deviation is 0.33 points, showing that the volatility of TT is relatively low. For Public Service Delivery (CU), the mean value of this variable is 6.98 points, and the standard deviation is 0.36 points, indicating a relatively low level of CU volatility. The fluctuation over the years of the components in the quality of public administration is presented in Figure 1.

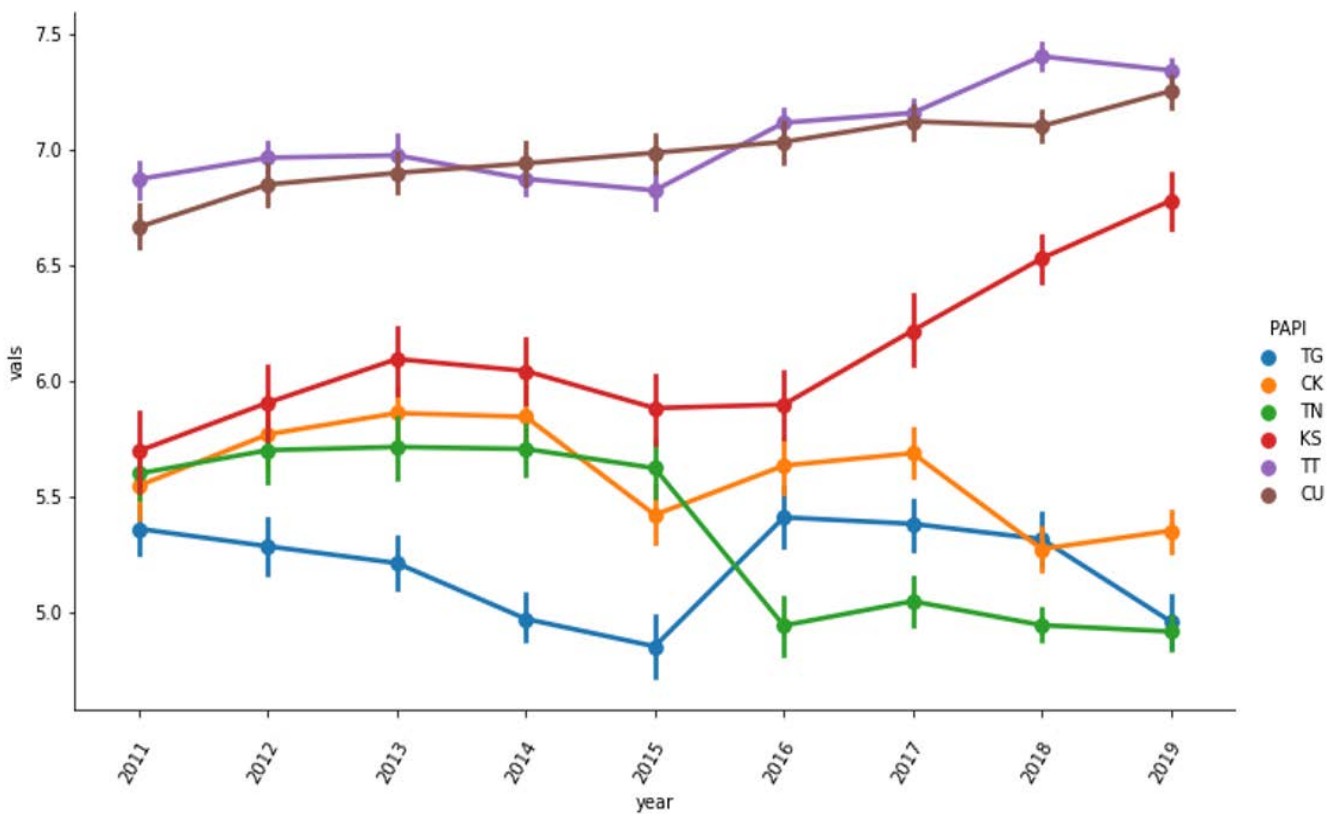

**Figure 1.** The change in the quality of public administration in the period 2011–2019.

### 4.2. Correlation Matrix

Figure 2 shows the correlation between each pair of variables in the research model.

To be able to measure the degree of the linear relationship between two variables, the correlation coefficient is used. The correlation coefficient, which represents the correlation between a pair of variables, has a value from −1 to 1. Figure 2 shows that the correlation coefficients of the pairs of independent variables are all less than 60%. Therefore, the independent variables in the model have a low correlation with each other. However, the correlation coefficients of the pairs of independent variables LNG and L.LNGRDP, LNK and L.LNGRDP, and LNK and LNG are higher than 60%.

Next, we conduct tests to ensure that there is no multicollinearity in the model. The results are presented in Table 3 below.

**Table 3.** Test multicollinearity of independent variables.

| Variables | VIF | 1/VIF |
| --- | --- | --- |
| lnGRDP | 4.50 | 0.222267 |
| lnK | 4.23 | 0.236201 |
| lnG | 2.91 | 0.343376 |
| CK | 2.06 | 0.485278 |
| TN | 1.57 | 0.635892 |
| lnL | 1.55 | 0.645772 |
| TG | 1.53 | 0.652205 |
| CU | 1.48 | 0.677167 |
| TT | 1.45 | 0.691525 |
| KS | 1.30 | 0.770655 |
| Average VIF | 2.26 | |

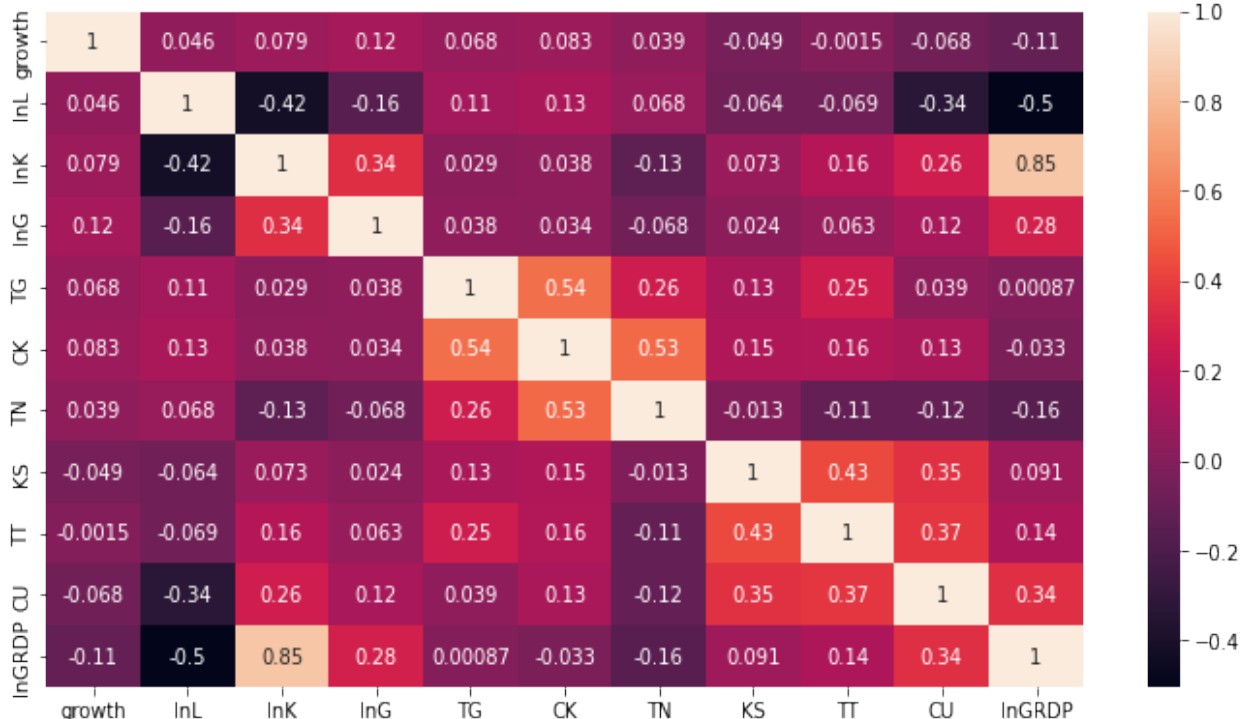

**Figure 2.** Correlation coefficient matrix.

The multicollinearity phenomenon is that the model's independent variables are expressed as a function and are linearly dependent on each other. According to Kleinbaum et al. (1988), there is high multicollinearity between the variables when the VIF index is greater than 5. The VIF test results as above show that the independent variables do not show high multicollinearity. Therefore, the authors use these variables for regression analysis.

Figure 3 shows that Control of Corruption in the Public Sector (KS), Public Administration Procedures (TT), and Public Service Delivery (CU) are positively correlated with the gross regional domestic product. However, Citizen Participation (TG), Publicity and Transparency in Decision Making (CK), and Citizens Accountability (TN) are not significantly correlated with the gross regional domestic product. However, this is just a visualization result. To be able to draw accurate conclusions about this relationship, we continue to estimate the model by the SGMM method.

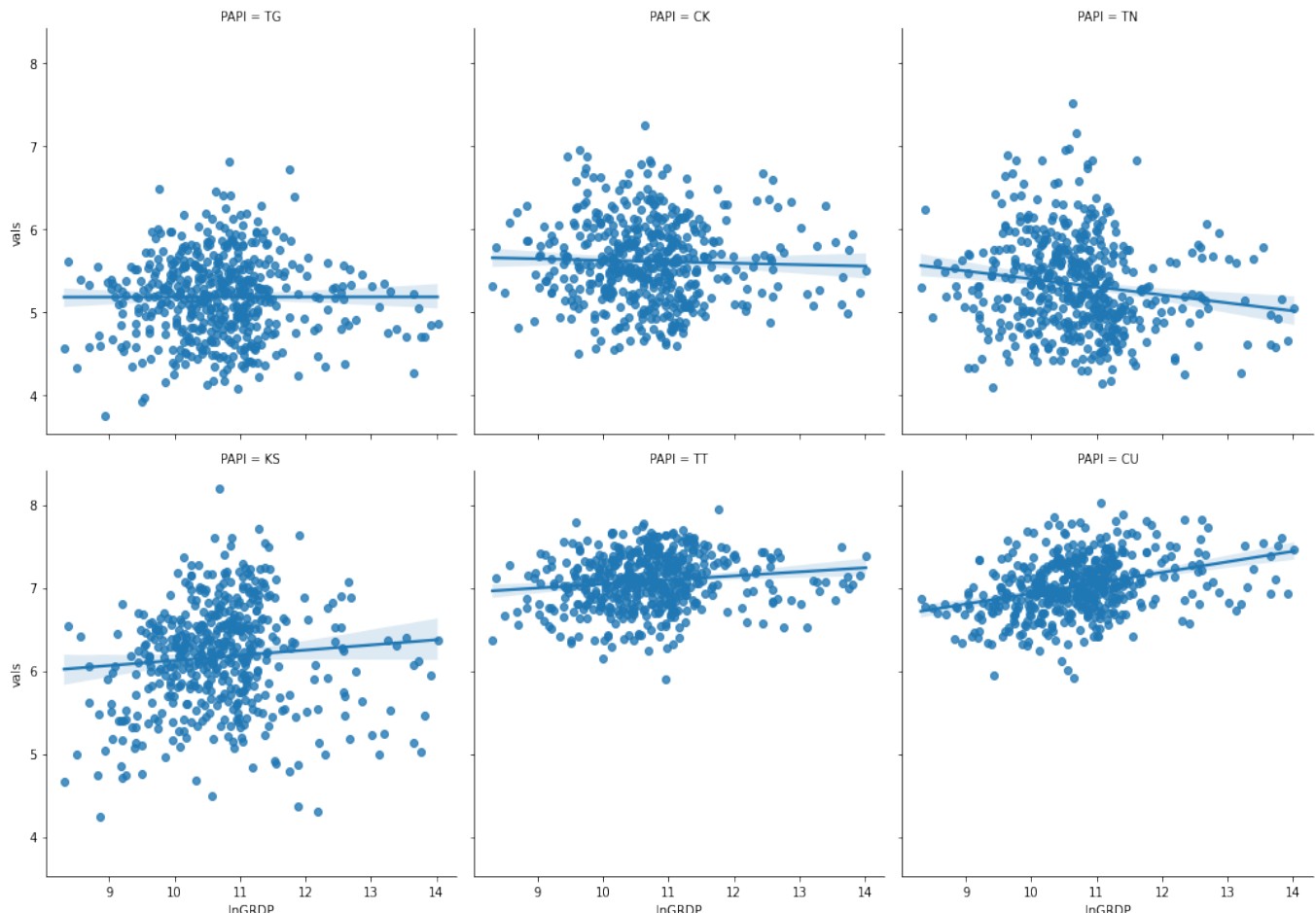

**Figure 3.** The relationship between the quality of public administration and the gross regional domestic product.

### 4.3. Estimation Result and Discussion

Table 4 shows the estimated results of the models that assess the impact of public administration quality on local economic growth in Vietnam.

Table 4 shows that the AR(1) test has a *p*-value less than the significance level of 10%, and the AR(2) test has a *p*-value greater than the significance level of 10%. Therefore, the model has first-order autocorrelation but no second-order autocorrelation with residuals. At the same time, the model has a *p*-value of Hansen's test greater than the significance level of 10%, which means there is a good fit for the instrumental variables used in the model. Besides, the F-test shows that the model is suitable when the *p*-value is less than the 1% significance level. At the same time, Table 4 is also satisfied that the number of instrumental variables must not exceed the number of observation groups. Therefore, the model has ensured reliability for us to continue the analysis.

Table 4 shows that there is an agreement with the effect of convergence when the regression coefficient of lnGRDP is significant at 1% and has a negative value. Specifically, provinces with small economic scale (small GRDP) have a higher growth rate than those with large economic scale (large GRDP) because the coefficient of lnGRDP carries a negative value. This result is consistent with neoclassical theory and findings in previous studies by Cooray (2009), Alexiou (2009), and Siddiqui and Ahmed (2013).

**Table 4.** Estimation result by the SGMM method.

| Growth | (1) | (2) | (3) | (4) | (5) | (6) |
|---|---|---|---|---|---|---|
| lnGRDP | −0.383 *** | −0.501 *** | −0.467 *** | −0.567 *** | −0.725 *** | −0.494 *** |
| lnG | 0.149 *** | 0.177 *** | 0.168 ** | 0.242 *** | 0.094 * | 0.173 ** |
| lnK | 0.223 ** | 0.364 *** | 0.284 ** | 0.377 ** | 0.636 *** | 0.333 ** |
| lnL | −0.788 | −0.947 | −2.053 * | −1.956 | −1.009 | −0.338 |
| TT | 0.073 ** | | | | | |
| TG | | 0.005 | | | | |
| KS | | | 0.079 *** | | | |
| CK | | | | −0.017 | | |
| TN | | | | | −0.021 | |
| CU | | | | | | 0.176 * |
| _CONS | 3.273 | 4.041 | 8.588 * | 8.223 | 4.909 * | 0.583 |
| AR (1) *p*-value | 0.084 | 0.091 | 0.085 | 0.091 | 0.004 | 0.088 |
| AR (2) *p*-value | 0.188 | 0.381 | 0.346 | 0.307 | 0.291 | 0.520 |
| Hansen *p*-value | 0.846 | 0.183 | 0.571 | 0.117 | 0.469 | 0.522 |
| Number of groups | 61 | 61 | 61 | 61 | 61 | 61 |
| Number of instruments | 12 | 11 | 13 | 13 | 12 | 12 |
| Second stage F-test *p*-value | 0.000 | 0.000 | 0.000 | 0.000 | 0.000 | 0.000 |

Estimation results of the impact of quality of public administration on local economic growth in Vietnam by SGMM method. Tests for the first-order and second-order correlation of residuals using AR (1), AR (2) *p*-value. Hansen's test for the appropriateness of instrumental variables through *p*-value. The F-test is used by the second stage F-test *p*-value for the fit of the model. *** Significance level of 1%; ** Significance level of 5%; * Significance level of 10%.

For local government expenditure, the regression coefficient of lnG is mostly significant at 10% and has positive values. Therefore, it is shown that increasing local government expenditure will promote local economic growth. Specifically, the regression coefficients of lnG in the models in Table 4 range from 0.094 to 0.242. Thus, a 1% increase in local government expenditure can increase local economic growth from 0.094% to 0.242%. Besides, the regression coefficients of lnK are all positive and significant at the 10% level. This result shows that increasing private investment capital in the locality will promote local economic growth and is consistent with endogenous growth theory. These results are also consistent with neoclassical theory and findings in previous studies by Cooray (2009), Alexiou (2009), and Siddiqui and Ahmed (2013).

Regarding the impact of public administration quality on local economic growth, Table 4 shows that the regression coefficients of the variables, including Control of Corruption in the Public Sector (KS), Public Administration Procedures (TT), and Public Service Delivery (CU), all have positive values and are significant. Thus, it can be seen that increasing the quality of public administration will positively impact local economic growth in Vietnam. These results support the theory of public administration, theory of public choice, theory of political economy, and new institutional economic theory.

## 5. Conclusions and Policy Implication

### 5.1. Conclusions

After more than a decade of implementing administrative reform in Vietnam in many aspects, the government recently started to focus on administrative reform. These innovative efforts are aimed at eliminating bureaucratic, cumbersome, and inconvenient procedures for the people. Public Administration Procedures (TT) measures the effectiveness of some services and administrative procedures based on people's actual experience when conducting some procedures that can be considered important and common in people's lives. Table 4 shows that the coefficient corresponding to TT has a value of 0.073 and is significant at the 5% level. Thus, when the quality of public administration, measured by

Public Administration Procedures (TT), increases by 1 unit, the local economic growth rate can increase by 7.3%.

Next, Control of Corruption in the Public Sector (KS) is perhaps one of the biggest challenges in state administration and management today because corruption seems to be deeply ingrained in the public apparatus. The regression coefficient corresponding to KS has a value of 0.079 and is significant at the 1% level. Thus, when the quality of public administration, as measured by Control of Corruption in the Public Sector (KS), increases by 1 unit, the local economic growth rate can increase by 7.9%.

Finally, Public Service Delivery (CU) directly assesses the actual "product" of good governance. Public Service Delivery (CU) considers four components, showing the four most important public services for people: Public health (health insurance and quality of local hospitals); Public primary education (with indicators such as the overall quality of public primary schools and the distance from home to school); Basic infrastructure (electricity to the house, quality of roads near the house, garbage collection service, and quality of domestic water); Security and order (security and order in residential areas and the seriousness of certain types of crimes occurring in residential areas). The regression coefficient corresponding to CU has a value of 0.176 and is significant at the 10% level. Thus, when the quality of public administration, measured by Public Service Delivery (CU), increases by 1 unit, the local economic growth rate can increase by 17.6%.

### 5.2. Policy Implication

The results show that increasing the quality of public administration will have a positive impact on local economic growth in Vietnam through component indicators, including Control of Corruption in the Public Sector (KS), Public Administration Procedures (TT), and Public Service Delivery (CU).

Thereby, in order to improve the local economic growth in Vietnam, it is necessary to have a number of solutions as follows.

Firstly, transparency of local public administration activities is required. It is very necessary to publicize and transparently plan and implement local mechanisms and policies. People can discuss and debate openly and democratically, and mechanisms and policies are supplemented and completed. Therefore, mechanisms and policies will be close to reality, meeting the country's development requirements. In addition, transparency also helps smooth local law enforcement, overcoming the situation of the same regulation but different interpretations by state agencies. Transparency in public administration helps businesses access information, increasing equality of business opportunities for businesses of all economic sectors. Public and transparent administrative procedures will contribute to overcoming corruption, creating favorable conditions for the people and production and business activities of enterprises. Simultaneously, transparency in governance is also the basis for people and businesses to monitor the activities of civil servants and overcome corruption.

In order for public governance to be transparent, local governments need to develop and implement a mechanism to ensure citizens' right to access information about the activities of agencies, organizations, and units. Local authorities also need to complete and strictly implement regulations on spokespersons of state agencies. In addition, the development and promulgation of laws on access to information, with sanctions for violators, also need to be implemented by local authorities.

Secondly, anti-corruption in the public sector is required. Fighting against the alienation of power or the abuse of power by those in power has been a problem throughout the development of human history. This struggle can be carried out "from outside" the power-holding apparatus, through the people's reaction to the phenomena of law violation and the use of power not for the interests of the people. This creates a constant, continuous, and widespread control with different forms. Execution of control can also be exercised "from within" the power apparatus, through institutions set up by the state, but first by organizing the assignment of power in the most reasonable way. That is the core of the

theory of decentralization, whose core is division and regulation, and mutual control between branches of state power. Some recommendations can be mentioned as strengthening propaganda and dissemination of views, guidelines, and policies of the Party and the state's anti-corruption laws, and strengthening handling and effectively preventing corruption. It is important to ensure that 100% of cadres, civil servants, public employees, and employees thoroughly grasp the anti-corruption legislation, and to strengthen and disseminate the law on anti-corruption deeply and widely among the people.

Thirdly, simplifying administrative procedures is required. Streamlining public administrative procedures helps to improve administrative efficiency and Public Administration. This further improves performance of public administrative service management, increasing the level of people's satisfaction with the operation of the government apparatus. The opinions and recommendations of the people will be answered in detail and will be a condition for the Provincial People's Committee to rectify and overcome the limitations of the authorities at all levels in order to better serve the people. For any issues that have not been directly answered at the meeting, the Provincial People's Committee will hold regular consultations and will give written answers to the people. It is important to continue to promote the reform of administrative procedures under local management and build simple and transparent administrative procedures. In addition, local authorities need to increase the ratio of online settlements to the total number of administrative procedures and promote electronic payments. In addition, it would be beneficial to deploy essential online public services as assessed by the United Nations, integrated and provided on the National Public Service portal. Local governments also need to step up the pilot implementation of the inter-connected model in handling a number of administrative procedures under the ministry's jurisdiction and strengthen the application of information technology in handling administrative procedures.

*5.3. Limitation and Suggestions*

Although the research model based on the production function and related studies has been developed, local economic growth can still be affected by other variables. Therefore, subsequent studies may add new variables to search for new results.

**Author Contributions:** T.T.H.H. conceived the idea and wrote the introduction. H.A.L. wrote the literature review and methodology. T.H.P. wrote the empirical results and conclusion. E.I.T. revised the manuscript. All authors have read and agreed to the published version of the manuscript.

**Funding:** This research received no external funding.

**Institutional Review Board Statement:** Not applicable.

**Informed Consent Statement:** Not applicable.

**Data Availability Statement:** Not applicable.

**Conflicts of Interest:** The authors declare no conflict of interest.

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
