# Peer review of "The Impact of Quality of Public Administration on Local Economic Growth in Vietnam"

_jrfm, doi:10.3390/jrfm15040158_

Round 1
Reviewer 1 Report
Comments
THE IMPACT OF QUALITY OF PUBLIC ADMINISTRATION ON LOCAL ECONOMIC GROWTH IN VIETNAM
This study examines how the quality of public administration influences local economic growth in Vietnam from 2011 to 2019. They used a Cobb-Douglas function examining both the individual and dual effects of state budget expenditures and quality of public administration on local economic growth in Vietnam. The system GMM method (SGMM) was used to estimate the model with data collected from 61 provinces and cities in Vietnam in the period 2011 - 2019. The findings suggest that state budget expenditures, quality of public administration positively influence local economic growth in Vietnam. Thereby, the authors propose policy implications to improve the efficiency of state budget expenditures on local economic growth in Vietnam in terms of public administration.
Although I enjoyed reading this paper, I think that several important adjustments are required in order to be published, as follows:
- The authors should be better motivated. In addition, the literature review is limited. I recommend the authors extend the literature review. The authors should enrich the literature review with relevant papers such as Faruq (2011), Konstantakopoulou (2022).
- What is a methodological innovations in their studies? How do they overcome the methodological issue in the previous research?
- The theoretical transmission channel 'quality of public administration- local economic growth' is not very clear, it being the core of empirical study. An extended discussion should be done here from my point of view.
- Authors should put more effort and thoroughly discuss point estimates, estimated effects, and the intuition behind the results backed up by the literature.
- The period of the paper sample is short, resulting in the undermining of the robustness of the estimated results.
References
Faruq, H.A., 2011. How institutions affect export quality. Economic Systems, 35 (4) 586-606.
Konstantakopoulou, I., 2022. Does health quality affect tourism? Evidence from system GMM estimates. Economic Analysis and Policy, 73, 425-440.
Reviewer 2 Report
Suggestions for improvements:
- In the introduction part, there are no research questions hypotheses, which is rather a serious flaw,
- please explain why those particular variables were selected.
Reviewer 3 Report
The paper could benefit from a structural evaluation of the concept of public administration from the Western perspective and compared with the Eastern perspective, especially by highlighting the main differences. This would be a great paragraph to insert in the literature review or in the methodology section.
Furthermore, to clear the approach on the subject the authors could insert a paragraph that underlines what technical purpose the paper has and what are the steps in the presented research. In the end of the research, you could insert a paragraph regarding future research or limits of the used model and methodology.
